# PHANTOM-DATA: TOWARDS A GENERAL SUBJECT-CONSISTENT VIDEO GENERATION DATASET

**Zhuowei Chen**[*], **Bingchuan Li**[*†], **Tianxiang Ma**[*], **Lijie Liu**[*], **Mingcong Liu, Yunsheng Jiang, Gen Li, Xinghui Li, Liyang Chen, Siyu Zhou, Qian He, Xinglong Wu**
Bytedance, China

## ABSTRACT

Subject-to-video generation has witnessed substantial progress in recent years. However, existing models still face significant challenges in faithfully following textual instructions. This limitation, commonly known as the copy-paste problem, arises from the widely used in-pair training paradigm. This approach inherently entangles subject identity with background and contextual attributes by sampling reference images from the same scene as the target video. To address this issue, we introduce **Phantom-Data, the first general-purpose cross-pair subject-to-video consistency dataset**, containing approximately one million identity-consistent pairs across diverse categories. Our dataset is constructed via a three-stage pipeline: (1) a general and input-aligned subject detection module, (2) large-scale cross-context subject retrieval from more than 53 million videos and 3 billion images, and (3) prior-guided identity verification to ensure visual consistency under contextual variation. Comprehensive experiments show that training with Phantom-Data significantly improves prompt alignment and visual quality while preserving identity consistency on par with in-pair baselines. Project Page:https://phantom-video.github.io/Phantom-Data/

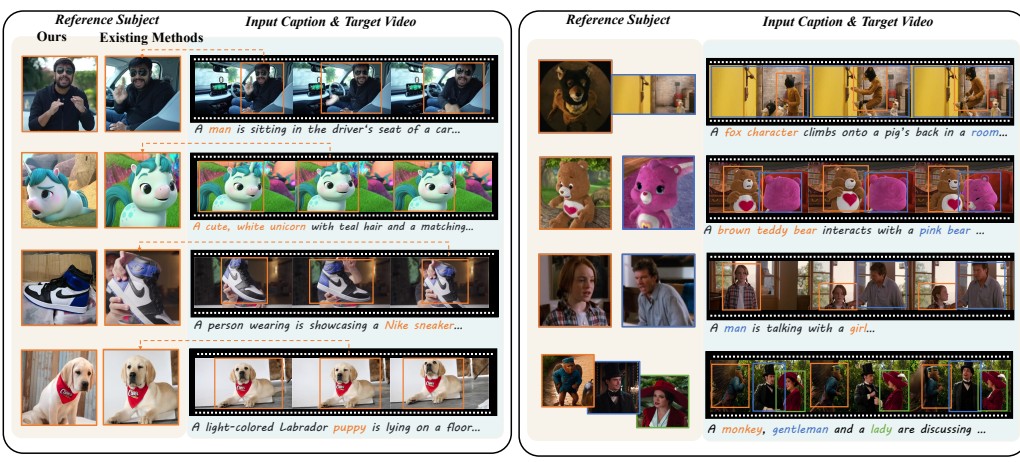

(a) Comparison on single-reference training data      (b) Multi-reference training data

Figure 1: Overview of training samples. (a) Single-reference setting: existing methods typically extract the reference image from the target video itself. In contrast, our approach uses reference images captured in distinct contexts. (b) Our dataset also includes multi-reference samples, presenting each subject in varied contextual settings.

## 1 INTRODUCTION

In recent years, text-to-video generation models, exemplified by Sora Brooks et al. (2024), have made significant progress Seawead et al. (2025); Kong et al. (2024); Wang et al. (2025a); Yang et al.

---

[*]Equal contributions, † Project lead

(2024b); Sand-AI (2025). However, due to the limited controllability inherent in textual instructions, achieving fine-grained control over video generation remains a key challenge for practical applications. Among recent advances Hu (2024); Jiang et al. (2025); Zeng et al. (2024); Feng et al. (2025); Kondratyuk et al. (2024); Xu et al. (2024), increasing attention has been paid to enforcing subject identity consistency in text-to-video generation. The subject-consistent video generation task (S2V) Liu et al. (2025b); Chen et al. (2025b); Huang et al. (2025) aims to generate videos that not only follow the given text prompt but also faithfully preserve the identity of reference subjects, such as people, animals, products, or scenes. This capability has great potential in applications such as personalized advertising Chen et al. (2025a) and AI-driven filmmaking Wu et al. (2025).

Despite encouraging progress in visual consistency, existing S2V approaches still suffer from limited text-following ability and suboptimal video quality, a phenomenon often referred to as the *copy-paste* problem. As shown in Fig. 2, the generated video directly replicates the reference subject from one of its frames, leading to the omission of the "boxing ring" background described in the prompt. This issue stems from the *in-pair* training paradigm, where the reference subject is sampled from the same target video, as illustrated in Fig.1(a). Consequently, the model tends to preserve not only subject identity but also irrelevant contextual details. However, in real-world scenarios, such entangled features may contradict the actions or semantics described in the text prompt, causing the generated videos that either deviate from the prompt or exhibit noticeable artifacts.

To address the above issue, prior works Chen et al. (2025b); Liang et al. (2025); Huang et al. (2025); Jiang et al. (2025); Ju et al. (2025) have explored various data normalization and augmentation strategies, such as background removal, color jittering, and geometric transformations. However, these methods struggle to unravel complex contextual factors, such as viewpoint and motion, due to limited variation. More recent approaches introduce *cross-pair* data, where identity-consistent reference and target frames are sampled from different sources. This setting encourages the model to focus on identity preservation while reducing overfitting to irrelevant visual contexts Polyak et al. (2024); Zhong et al. (2025). However, existing cross-pair datasets are primarily limited to facial domains, making them difficult to generalize to general subject scenarios. Overall, current training datasets either provide insufficient reference variation or lack domain diversity, limiting the effectiveness of easing copy-paste problem.

In this work, we introduce *Phantom-Data*, a subject-to-video dataset specifically constructed to mitigate the prevalent copy-paste problem in the general scenarios. It is built around three core design principles for the reference subject: 1) **General and input aligned subjects**: Reference images should span a wide range of commonly encountered subject types and reflect the distribution of real-world user inputs. 2) **Different contexts**: Reference subjects appear in varied conditions—such as different backgrounds, viewpoints, or poses—relative to their counterparts in the target video. This encourages the model to generalize identity preservation under distribution shifts and reduces reliance on spurious identity-irrelevant correlations. 3) **Consistent identity**: Despite contextual variation, the reference subject must remain visually consistent with the target video subject in terms of shape, structure, and texture.

To fulfill these principles, we design a three-stage pipeline: Firstly, we perform **S2V Detection** by leveraging a vision-language model to conduct open-set object detection and identify candidate subjects of appropriate size. A second-stage filtering step further refines the results by retaining only subjects that are both semantically relevant and visually compact. Then, we conduct **Contextually Diverse Retrieval** by constructing a large-scale subject database comprising over 53 million video segments and 3 billion image samples, increasing the likelihood of retrieving the same identity under diverse backgrounds, poses, and viewpoints. Finally, we apply **Prior-Guided Identity Verification** to ensure identity consistency. For living beings (e.g., humans, animals), we mine temporal structures from long videos to construct cross-context pairs. For static objects (e.g., products), we perform category-specific retrieval. A final VLM-based pairwise check verifies that each selected pair maintains both identity consistency and contextual diversity. Through this pipeline, we construct a large-scale, high-quality cross-pair consistency dataset comprising approximately *1 million identity-consistent pairs* with over *30,000 multi-subject scenes*, offering a strong foundation for modeling general subject-to-video tasks. Representative samples are shown in Fig. 1.

To validate the effectiveness of our dataset, we conduct comprehensive experiments using open-source video generation models. The results demonstrate that, compared to prior data construction methods, our cross-pair approach substantially improves key metrics such as text alignment and vi-

*Reference Subject*                                       *Input Caption & Target Video*

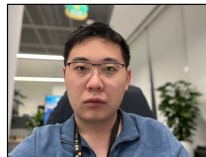 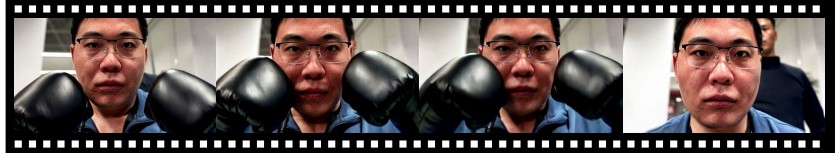

The man is boxing in the ***boxing ring***, waving his fists and sweating on his cheeks.

Figure 2: Illustration of the copy-paste problem. The shown result is generated by a SOTA video generation model (Kling Kli).

sual quality, while maintaining identity consistency on par with in-pair baselines. Furthermore, we perform ablation studies to highlight the importance of a large-scale and diverse cross-pair dataset, showing that both data volume and scene diversity play a critical role in enhancing generation performance. We also validate the effectiveness of our data pipeline in preserving high identity consistency while ensuring sufficient contextual diversity.

Our main contributions can be summarized as follows:

- We introduce Phantom-Data, the first general-purpose cross-pair video consistency dataset, comprising approximately 1 million high-quality, identity-consistent pairs that span a wide range of subject categories and visual contexts.

- We present a structured data-construction pipeline purpose-built for subject-consistent video generation. It unifies a subject-centric detection module optimized for the S2V task, large-scale cross-context retrieval, and prior-guided identity verification, thereby securing strict identity fidelity while introducing rich contextual diversity.

- We conduct extensive experiments to validate the effectiveness of our dataset, demonstrating consistent improvements in text alignment, visual quality, and generalization over existing in-pair baselines.

| Method | General Objects | Input-aligned Objects | Diverse Context | Publicly Available |
|---|---|---|---|---|
| MovieGen | ✗ | ✓ | ✓ | ✗ |
| Video Alchemist | ✓ | ✗ | ✗ | ✗ |
| ConceptMaster | ✓ | ✗ | ✗ | ✗ |
| Ours | ✓ | ✓ | ✓ | ✓ |

Table 1: Comparison between Phantom-data and datasets used in prior work.

## 1.1 RELATED WORK

**Text-to-Video Generation.** Early diffusion-based video generators Blattmann et al. (2023); Guo et al. (2024a); Wang et al. (2024b) were limited to producing short clips with constrained spatial and temporal resolution. However, the field has rapidly progressed with the introduction of large-scale latent diffusion models and transformer-based architectures. Notably, Sora Brooks et al. (2024) is capable of generating minute-long, high-fidelity videos, while contemporaneous systems such as Seaweed Seawead et al. (2025), Hunyuan-Video Kong et al. (2024), CogVideo-X Yang et al. (2024b), MAGI Sand-AI (2025), and others Wang et al. (2025a) have further advanced frame rate, resolution, scene complexity, realism, and motion smoothness. Despite their impressive visual quality, these generic text-conditioned models provide only coarse control: textual prompts alone cannot fully specify scene layout, subject appearance, or viewpoint, motivating research into finer control signals.

**Subject-Consistent Video Generation.** The task of subject-consistent video generation (S2V) Liu et al. (2025b); Fei et al. (2025); Chen et al. (2025b); Huang et al. (2025); Deng et al. (2025) focuses on generating videos that not only align with the given text prompt but also preserve the visual identity of a reference subject, such as a person, animal, product, or scene. From a modeling

perspective, one common strategy Huang et al. (2025); Chen et al. (2025b); Polyak et al. (2024); Hu et al. (2025) is cross-attention-based fusion, where visual features extracted from pretrained encodersRadford et al. (2021); Oquab et al. (2024); Xu et al. or VLMs, are injected into the generative backbone through dedicated attention layers. An alternative approach is noise-space conditioning, where identity features obtained from a VAE encoder are directly concatenated with the noise input of the diffusion model, without modifying the underlying architecture. This lightweight design enables nearly lossless injection of identity information, as seen in DIT-style models such as Phantom Liu et al. (2025b) and VACE Jiang et al. (2025). Recent systems like SkyReels-A2 Fei et al. (2025) explore combining both strategies, incorporating cross-attention guidance and noise-level conditioning within a unified framework.

**Training Data in Subject-to-Video Generation.** Training data plays a crucial role in subject-consistent video generation, as it directly influences a model's ability to generate faithful and controllable results. Most existing approaches rely on *in-pair* supervision, where the reference and target frames are sampled from the same video clip. While this setup guarantees identity alignment, it often leads to the undesirable *copy-paste* effect—where the model reproduces not only the subject but also the background and pose of the reference frame, limiting its capacity to follow the input prompt. To mitigate this issue, several works Chen et al. (2025b); Liang et al. (2025); Huang et al. (2025); Jiang et al. (2025); Ju et al. (2025) adopt data normalization and augmentation strategies, such as background removal, color jittering, and geometric transformations. However, these techniques, combined with the limited diversity inherent in in-pair training, are often insufficient to address complex contextual variations such as motion, viewpoint, and scene layout. Recent efforts have turned to *cross-pair* training, where identity-consistent reference and target frames are sampled from different videos. This setting encourages the model to concentrate on subject identity while reducing overfitting to specific visual contexts Polyak et al. (2024); Zhong et al. (2025). Nevertheless, current cross-pair datasets are mostly restricted to narrow domains like human faces, limiting their generalizability to broader subject categories such as animals, products, or stylized characters. In summary, although cross-pair supervision offers a promising direction for addressing the copy-paste issue, the absence of high-quality, diverse, and identity-consistent training data across general domains remains a significant bottleneck for advancing S2V models. To bridge this gap, we introduce Phantom-data, a large-scale cross-pair dataset designed to support subject-consistent video generation across a wide range of real-world categories.

## 2 PHANTOM DATA

We provide a detailed analysis of **Phantom-Data**, focusing on its statistical properties and a comparison with existing datasets for subject-consistent video generation.

### 2.1 STATISTICAL ANALYSIS

We analyze the dataset at both the video and subject levels.

**Video-level properties.** As shown in Fig. 3(a–c), our dataset spans a wide range of video durations, resolutions, and motion patterns. Around 50% of videos are 5–10 seconds long, and the majority are in 720p resolution. Motion levels also vary considerably, covering both relatively static and highly dynamic scenes.

**Subject composition.** Fig. 3(d) illustrates the distribution of subject types and their combinations. While the majority of samples (approximately 720,000) contain a single subject—such as a human, product, or animal—a substantial portion (around 280,000) involve two or more co-occurring entities, supporting multi-subject consistency modeling.

**Reference diversity.** As shown in Fig. 3(e), the dataset spans a broad semantic space of subject categories. Common reference entities include humans (e.g., woman, man, girl), animals (e.g., dog, bird), and man-made objects (e.g., smartphone, car, laptop), highlighting the dataset's suitability for general-purpose subject-to-video modeling across varied domains.

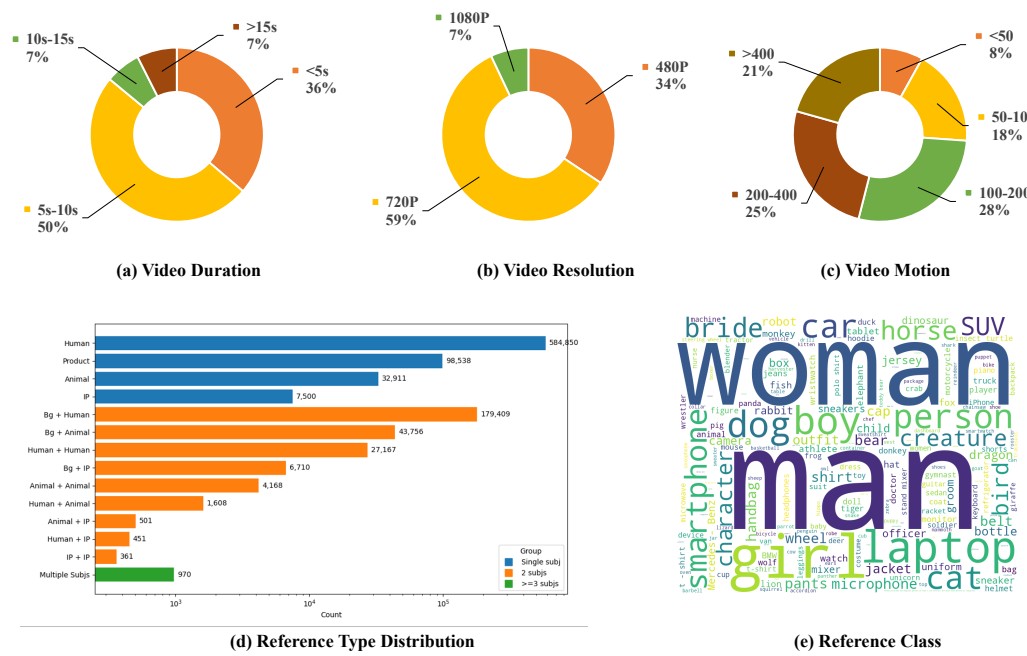

Figure 3: The statistical analysis of Phantom-Data.

## 2.2 COMPARISON WITH PRIOR DATASETS

As summarized in Table 1, existing datasets for subject-consistent video generation either lack general object coverage, rely heavily on input-aligned references from the same video, or are limited in contextual diversity. In contrast, Phantom-Data offers a more comprehensive setting: it supports general object categories beyond faces, encourages cross-context modeling by sampling subject-reference pairs from diverse scenes, and is publicly available for research. This makes it the first open-access dataset to jointly support identity consistency and context diversity in a general-purpose, cross-pair setup.

## 3 DATA PIPELINE

### 3.1 VIDEO DATA SOURCE

The Phantom-Data video dataset consists of clips collected from public sources such as Koala-36M Wang et al. (2024a), as well as proprietary internal repositories. Each video undergoes a rigorous quality control pipeline, including black border detection, motion analysis, and other filtering steps. Subsequently, long videos are segmented into short clips at the second level using scene segmentation. Each resulting clip is then annotated with a corresponding video caption. The total number of videos is approximately 53 million.

### 3.2 DATA PIPELINE

Given an input video and its associated caption, we focus on constructing a high-quality *cross-pair* dataset, where the same subject appears across different visual contexts while maintaining identity consistency. To this end, we design a structured data pipeline consisting of three key stages. As shown in Fig.4, firstly, we perform *S2V Detection* to identify high-quality subject instances from videos. Then, we propose a *Contextually Diverse Retrieval* module to recall candidate images that are likely to correspond to the detected subjects across varying scenes. Finally, we apply *Prior-based Identity Verification* to filter the retrieved candidates, ensuring that only those sharing the same identity across different contexts are retained.

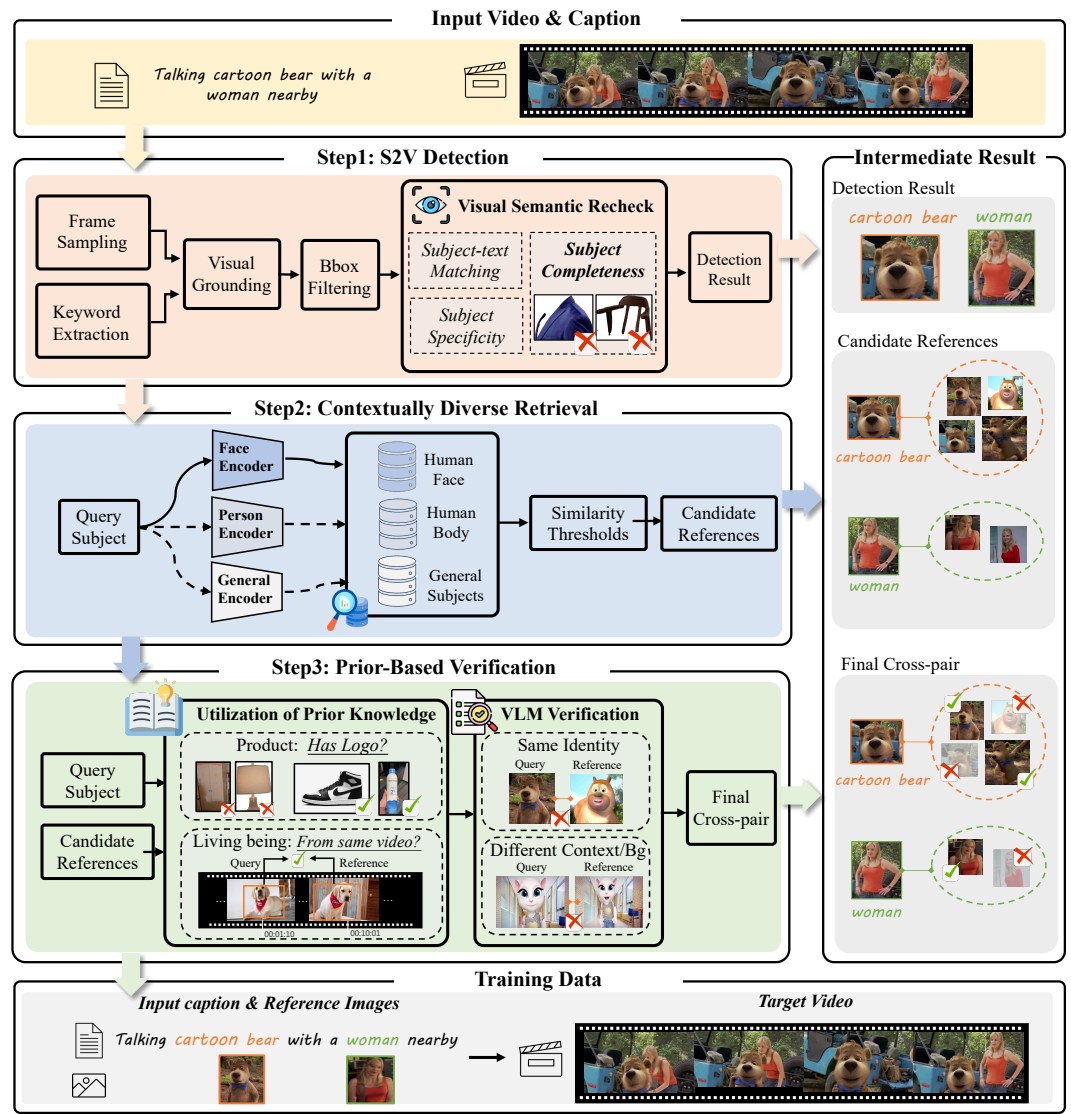

Figure 4: The overview of the data pipeline for constructing cross-pair training samples.

### 3.2.1 S2V DETECTION

This stage aims to identify diverse and qualified subjects from each video clip as candidates for cross-scene pairing. It consists of five major steps:

**1. Frame Sampling** To reduce computation, we sample three frames at $t = 0.05$, 0.5, and 0.95 of each clip, following Chen et al. (2025b), ensuring temporal diversity.

**2. Keyword Extraction.** We use Qwen2.5 Yang et al. (2024a) to extract key noun phrases (e.g., people, animals, products) from captions, serving as subject candidates for grounding.

**3. Visual Grounding.** Qwen2.5-VL Bai et al. (2025) aligns each phrase to regions in the sampled frames. Ambiguous matches mapping to multiple regions are removed to reduce noise.

**4. Bbox Filtering.** We retain boxes covering between $4\%$ and $90\%$ of the image and at least $128 \times 128$ in size. Overlapping boxes (IoU $> 0.8$) are suppressed for clarity.

**5. Visual-Semantic Recheck.** To further ensure the quality of the grounded subjects, we employ another vision-language model, InternVL2.5 7B Chen et al. (2024), to validate each detection against

the following criteria: 1) Completeness: We observe that visual grounding often produces bounding boxes around partial or cropped objects, as a result of the underlying detection model's exhaustive labeling strategy. However, for S2V task, users typically provide complete reference subjects, making such incomplete detections unsuitable. We therefore filter out any region that fails to cover the full extent of the object. 2) Specificity: The subject must be visually distinct and identifiable. Vague or generic objects, such as trees, rocks, or background clutter, are excluded. 3) Subject-text Matching: The grounded region must be semantically consistent with the associated phrase. To improve alignment precision, we employ a separate instance of InternVL2.5 to reevaluate the consistency between the textual description and the detected subject.

As a result of this pipeline, we obtain a high-quality set of subject instances, each paired with a corresponding descriptive phrase. Since a subject may appear in multiple frames across the video, we select only one representative instance for visualization in the Intermediate Result of Fig. 4.

### 3.2.2 CONTEXTUALLY DIVERSE RETRIEVAL

Given the subject instances detected in the previous stage, we aim to find candidate reference images of the same subject appearing in different visual contexts. To achieve this, we construct a large-scale retrieval bank and use the detected subjects to perform identity-aware querying.

**Large-Scale Retrieval Bank Construction.** The retrieval bank comprises two essential components: diverse subject image sources to increase contextual variability, and feature representations tailored for identity-preserving retrieval.

*Subject Source.* We begin by registering every detected subject instance from the training videos into a retrieval bank. To further broaden candidate diversity, we augment this bank with an extra 3 billion images from the LAION dataset Schuhmann et al. (2022) beyond the original video corpus. These external images inject greater variation in scene, pose, and appearance, delivering broader contextual coverage during retrieval, an advantage that is particularly valuable for product-centric scenarios with substantial intra-instance variation.

*Subject Representation.* To support reliable cross-context identity matching, we employ expert-designed encoders to extract identity-preserving and context-invariant embeddings tailored to different subject categories. These embeddings are used for both indexing the retrieval bank and querying.

For facial representation, we adopt the widely used ArcFace encoder Deng et al. (2019) to extract robust and discriminative identity embeddings:

$$V_{\text{face}} = E_{\text{arcface}}(I_{\text{face}}). \tag{1}$$

For general objects, inspired by ObjectMate Winter et al. (2024), we employ a CLIP-based model fine-tuned on a consistency-focused image dataset Shao and Cui (2023) to extract identity-preserving embeddings:

$$V_{\text{subj}} = E_{\text{IR}}(I). \tag{2}$$

For human subjects, which are central to many downstream applications, we combine both facial and clothing features. Each individual is represented by concatenating the general appearance embedding with the corresponding facial embedding:

$$V_{\text{person}} = [E_{\text{IR}}(I), E_{\text{arcface}}(I_{\text{face}})]. \tag{3}$$

**Query-Based Retrieval.** To ensure the retrieved candidates are visually distinct from the query image yet share the same identity, we apply both upper and lower bounds on similarity. Specifically, we discard overly similar results (potential duplicates) by enforcing an upper similarity threshold, and exclude unrelated identities by applying a lower threshold.

### 3.2.3 PRIOR-BASED IDENTITY VERIFICATION

However, due to the large scale of the retrieval corpus, false positives frequently occur even within seemingly reasonable similarity ranges. To address this issue, we adopt a two-stage filtering strategy based on prior knowledge and VLM Verification.

| Methods | Subject Consistency | | Prompt Following | Video Quality | | | | |
|---------|-----------|-----------|-----------|-----------|-----------|-----------|-----------|-----------|
| | DINO ↑ | GPT-4o ↑ | Reward-TA ↑ | Temporal ↑ | Motion ↑ | IQ ↑ | BG ↑ | Subj ↑ |
| In-pair | **0.478** | 2.481 | 2.074 | 0.971 | 0.985 | 0.725 | 0.937 | 0.933 |
| In-pair + Data Aug | 0.473 | 2.792 | 2.427 | 0.961 | 0.979 | 0.730 | 0.932 | 0.922 |
| Face Cross-pair | 0.354 | 2.378 | 3.022 | **0.983** | **0.989** | 0.723 | 0.937 | 0.935 |
| Ours | 0.416 | **3.041** | **3.827** | 0.975 | 0.986 | **0.739** | **0.948** | **0.944** |

Table 2: Main results comparing prompt following, subject consistency, and video quality across different training paradigms. Bold denotes the best performance per column. The underline indicates the second-highest scores.

**Utilization of Prior Knowledge.** We apply category-specific filtering strategies to improve cross-pair reliability: 1) *Non-living subjects* (e.g., products): These typically exhibit high intra-class variability, making identity verification more challenging. To improve precision, we retain only product instances that feature complete and recognizable brand logos (e.g., Nike, Audi), which remain visible across different scenes. 2) *Living entities* (e.g., humans, animals): For these subjects, we restrict retrieved candidates to those from different clips within the same long-form video. This constraint ensures natural variation in scene and pose while maintaining consistent identity.

**VLM-Based Consistency Verification.** To further ensure both identity consistency and contextual diversity, we apply a VLM-based verification procedure: 1) *identity consistency*: For non-living objects, we enforce strict similarity in visual details such as color, packaging, and textual elements, while allowing for background variation. For living subjects, especially humans, we verify facial identity consistency and, in the case of full-body samples, also ensure clothing alignment. 2) *Contextual diversity*. We keep only those cross-pair samples that exhibit substantial variation in background and scene context, thereby alleviating copy-paste artifacts during model training.

## 4 EXPERIMENTS

### 4.1 IMPLEMENTATION

**Model Architecture.** We validate the effectiveness of our proposed data using the Phantom-wan Liu et al. (2025b) model. Built on the Wan2.1 Wang et al. (2025b) foundation, Phantom-wan is a leading open-source framework for subject-consistent video generation.

**Training and inference.** We train a 1.3 billion-parameter Phantom-wan model using Rectified Flow (RF) Lipman et al. (2022); Liu et al. as the training objective. The training is performed on 64 A100 GPUs for 30k iterations with 480p resolution data, which yields stable performance. During inference, we apply Euler sampling with 50 steps and use classifier-free guidance Ho and Salimans (2022) to decouple image and text conditions. All experiments follow the same training and inference settings to ensure fair comparisons.

**Evaluation.** We construct a test suite of 100 cases from diverse scenarios, covering humans, animals, products, environments, and clothing. These cases include both single- and multi-subject settings, paired with manually written text prompts that reflect natural user input.

We evaluate model performance across three dimensions: video quality, text-video consistency, and subject-video consistency. Subject-video consistency is evaluated using CLIP Guo et al. (2024b), DINO Oquab et al. (2023), and GPT-4o scores, following recent evaluation protocols inspired by Peng et al. (2024). Text-video consistency is measured via Reward-TA Liu et al. (2025a), both of which assess the semantic alignment between generated video content and the text prompt. Video quality is assessed using VBench Huang et al. (2024), which provide fine-grained evaluation along several aspects, including Temporal (temporal flickering and stability), Motion (smoothness of subject motion), IQ (overall imaging quality), BG (background consistency across frames), and Subj (temporal consistency of the generated subject).

### 4.2 MAIN RESULTS

We evaluate our method against three representative baselines: (1) In-pair training, which samples the reference subject from the same video; (2) In-pair with copy-augmentation, which introduces

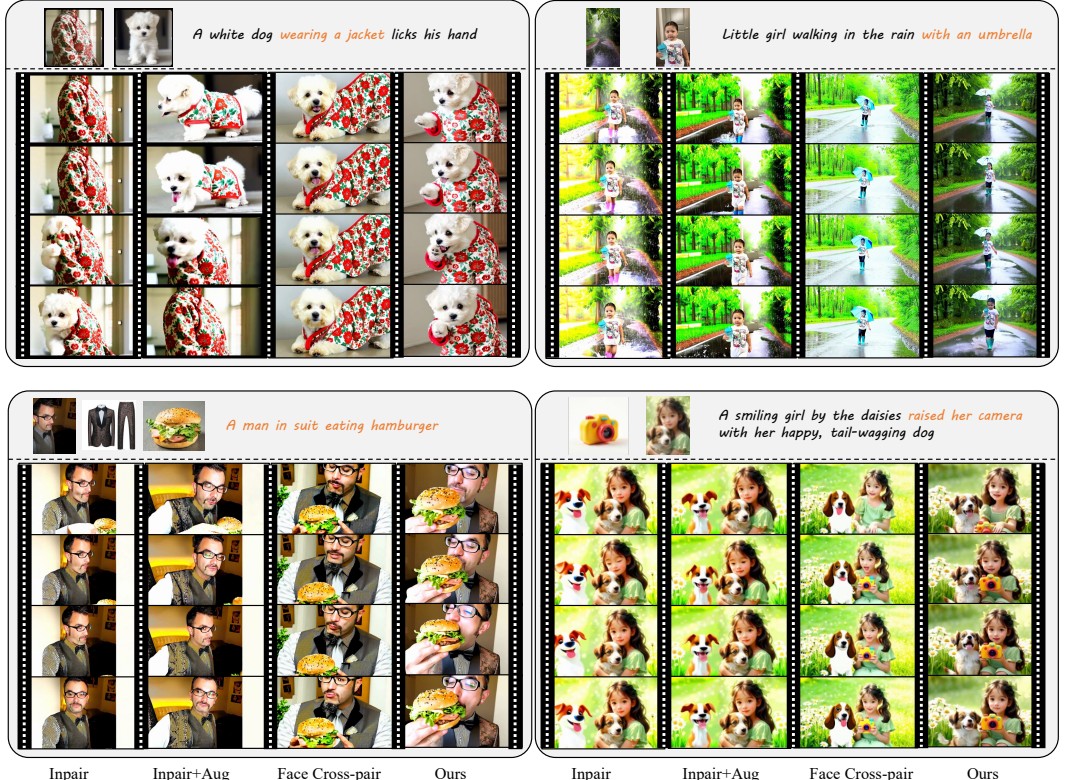

Figure 5: Qualitative comparisons across different training strategies.

spatial and appearance augmentations to reduce overfitting as in Chen et al. (2025b); and (3) Face-based cross-pair, which utilizes face-level identity matching across videos. We report both quantitative and qualitative comparisons.

Quantitative results demonstrate that our cross-pair training paradigm achieves state-of-the-art performance in terms of text-video alignment and overall video quality, as measured by reward-based evaluation metrics. Furthermore, our method delivers competitive subject consistency, rivaling in-pair baselines despite the increased scene diversity. In contrast, In-pair settings suffer from poor text-following ability due to overfitting on narrow visual contexts. The Face cross-pair method performs slightly better on prompt following but is limited by the narrow domain of its face-centric training data, resulting in weaker identity preservation across diverse subjects.

Qualitative comparisons, as shown in Fig.5, further support our conclusions. Across multiple prompts and subject categories, models trained with in-pair data consistently fail to follow textual instructions, often generating videos with obvious artifacts. In contrast, our cross-pair trained model successfully aligns with the prompt across all cases, producing coherent and faithful subject-driven videos.

### 4.3 Ablation Studies

**Subject Diversity.** As shown in Table 3, enriching the training set with diverse subject types—including humans, animals, products, and multi-subject scenes—consistently improves subject consistency and prompt following, compared to the face-only baseline.

**Data Scale.** Table 4 illustrates the effect of data scale. Increasing the training set from 100 k to 1 million samples leads to further improvements across all metrics, highlighting the importance of both diversity and scale in building a robust subject-to-video generation dataset.

**Contextually Diverse Retrieval.** To assess the impact of contextual diversity in reference selection, we compare different sampling and retrieval strategies: (1) Temporal Sampling. As illustrated in

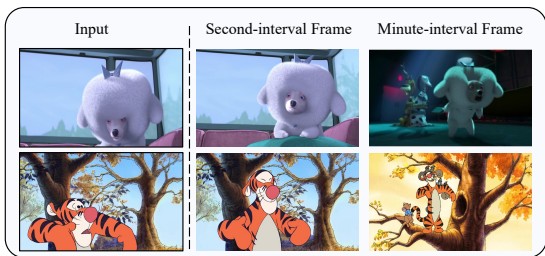

(a) Reference frames sampled at different temporal intervals

(b) Retrieved top-1(R@1) results from different source datasets

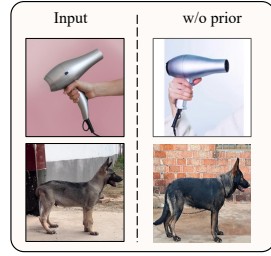

(c) False positive result if prior is not used

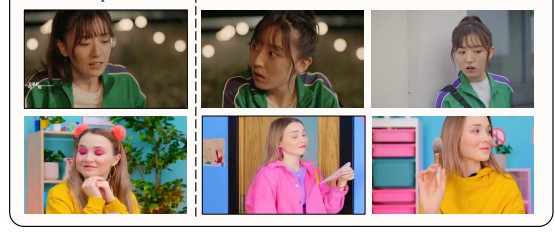

(d) False positive results on person class (same face id +same cloth)

Figure 6: Ablation study on Contextually Diverse Retrieval and Prior-Based Identity Verification. (a) Reference frames from different timestamps show that longer videos offer more diverse contexts. (b) Retrieval from large-scale image datasets improves recall and candidate diversity. (c) Without prior filtering, false positives may be included. (d) Verification removes mismatched or overly similar identities, ensuring high-quality pairs.

Table 3: Ablation study on subject diversity.

| Methods | Subject Consistency | | Prompt Following |
|---|---|---|---|
| | DINO ↑ | GPT-4o ↑ | reward-TA ↑ |
| baseline (face only) | 0.354 | 2.378 | 3.022 |
| + human | $0.401_{+0.047}$ | $2.747_{+0.363}$ | $3.726_{+0.702}$ |
| + IP/animal | $0.416_{+0.062}$ | $2.795_{+0.411}$ | $3.407_{+0.383}$ |
| + product | $0.386_{+0.032}$ | $2.662_{+0.288}$ | $3.572_{+0.58}$ |
| + multi-subject | $0.418_{+0.064}$ | $2.901_{+0.525}$ | $3.512_{+0.498}$ |

Table 4: Ablation study on data scale.

| Methods | Subject Consistency | | Prompt Following |
|---|---|---|---|
| | DINO ↑ | GPT-4o ↑ | reward-TA ↑ |
| 100k | 0.408 | 3.090 | 3.796 |
| 1 M | **0.416** | **3.175** | **3.827** |

Fig. 6(a), reference frames sampled at longer temporal intervals (e.g., minute-level vs. second-level) provides richer visual diversity. (2) Multi-source Retrieval. Fig. 6(b) compares retrieval from video-only sources and from a combined image+video retrieval bank. Incorporating large-scale image datasets improves both recall and candidate diversity.

**Prior-Based Identity Verification.** We further evaluate the role of prior filtering and identity verification in ensuring training quality: (1) Prior Filtering. Without prior-based constraints, visually similar but semantically incorrect matches (false positives) are often included (see Fig. 6(c)). (2) Verification Module. As shown in Fig. 6(d), our identity verification module further refines the candidate set by removing both near-duplicates (overly similar samples) and mismatched identities (overly dissimilar ones).

## 5 CONCLUSION

We propose Phantom-Data, a large-scale, general-purpose cross-pair dataset to improve subject consistency and text alignment in text-to-video generation. By introducing a structured pipeline, combining open-vocabulary detection, diverse cross-context retrieval, and identity verification, we address the limitations of in-pair training and reduce the copy-paste problem. Experiments show that our dataset significantly boosts generation quality while maintaining strong identity consistency. Phantom-Data provides a solid foundation for future research in controllable subject-to-video generation task. In future work, we will explore how to extend the reference image domain from specific subjects to general references, enabling users to create videos with flexible control over various elements through both images and text prompts.

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

## 6 Appendix

### 6.1 The Limitations of Synthetic Data

We try two SOTA models, GPT4o and DreamO Mou et al. (2025) to generate consistent subjects on different context. The results in Fig.7 shows these models could still generated inconsistent subject. However our real cross-pair data construction pipeline could provide exactly same subjects on different context.

### 6.2 User study

To evaluate the generated videos under different training data regimes, we conducted a user study comparing four settings: in-pair training, in-pair with data augmentation, face-level cross-pair training, and our proposed full-object cross-pair approach. We ask six participants, each of whom independently evaluated 50 video groups, containing four videos generated in the different training settings. For each group, participants were asked to select the best video in terms of overall visual quality, subject consistency, and alignment with textual prompts. As summarized in Table 5, our method was overwhelmingly preferred, receiving **76%** of the votes. In contrast, all other baselines received less than 12%, highlighting the effectiveness of our cross-pair training design in producing videos that are faithful to the intent of users.

| In-pair | In-pair + Data Aug | Face Cross-pair | Ours |
|---------|--------------------|-----------------|------|
| 6% | 11% | 7% | 76% |

Table 5: User study on the best video selection based on overall visual quality, subject consistency, and text alignment across different training data settings.

### 6.3 Prompts for Each Tasks

### 6.3.1 Keyword Extraction

We leverage the Qwen2.5 7B language model Yang et al. (2024a) to extract key noun phrases—such as persons, animals, products, and IP characters—from video captions. Given the inherent variability in LLM outputs, we incorporate a second-stage LLM for post-processing, which enforces structured JSON formatting and filters out spurious or noisy detections. To enhance phrase quality, we apply the following constraints:

1. **Structural Filtering**: Exclude generic body parts or structural terms (e.g., *leg*).
2. **Number Filtering**: Eliminate plural forms (e.g., *women*) to ensure instance-level grounding.
3. **Specificity Enforcement**: Promote fine-grained disambiguation within categories, encouraging phrases like "a woman with a red coat" or "a woman with red hair" rather than generic references.

| Input Image | Data Aug | DreamO | GPT4-o | Ours |
|---|---|---|---|---|

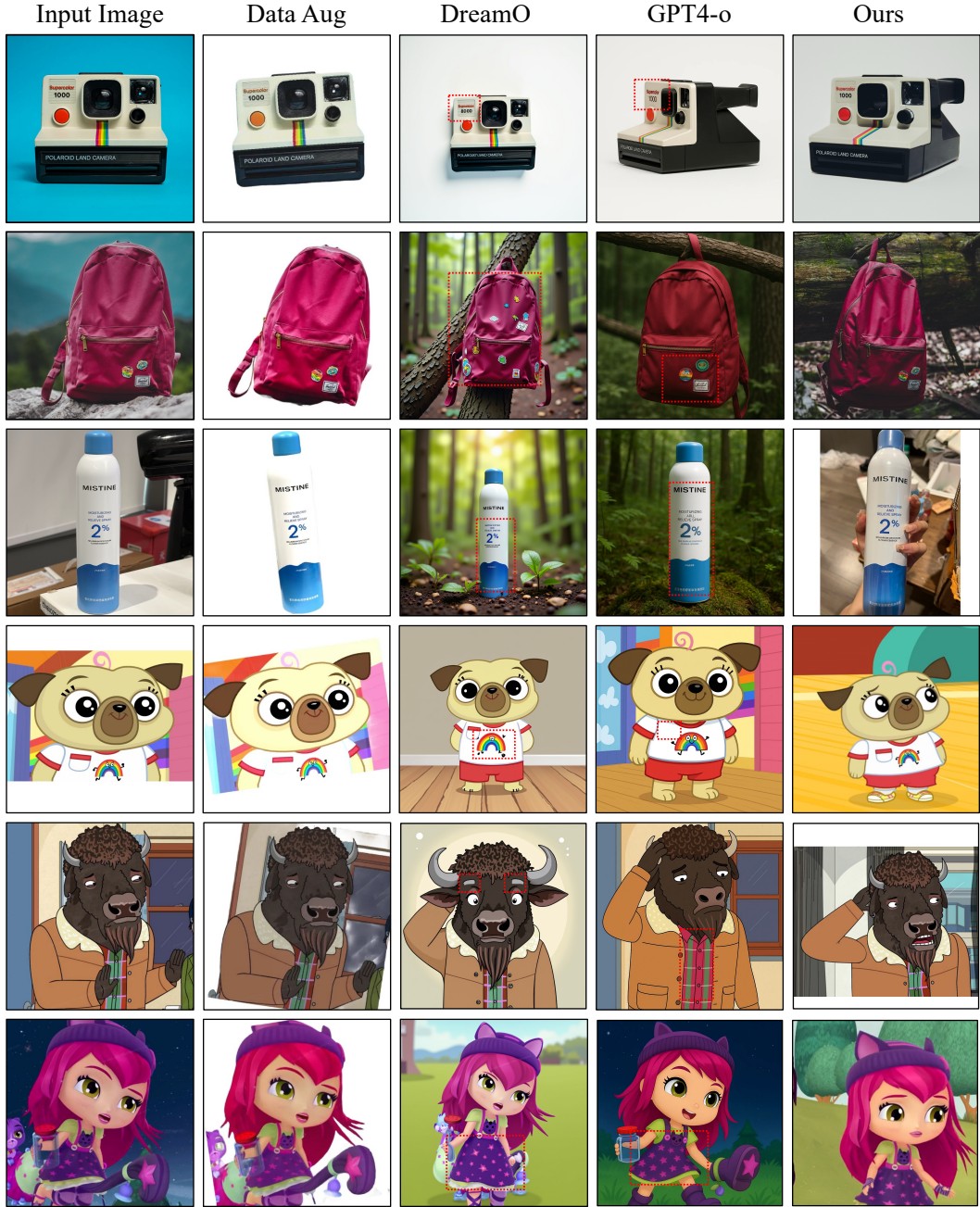

Figure 7: Comparison of reference-data construction strategies. Naïve data augmentation (Data Aug) offers only limited variation, while directly adopting state-of-the-art IP-consistent generators (e.g., DreamO, GPT-4o) can introduce appearance inconsistencies, highlighted in the figure. Best viewed in color with zoom.

**Keyword Extraction Prompts - First Round**

**User Prompt**

```
Given an video caption, please retrieve the entity words that
    indicate human and subjects.
[Definition of humans] human that appear in the image.
```

```
[Definition of subjects] belong to products(e.g. bike/cloth
    ...)/animal/IP Character
All entity words need to strictly follow two rules below:
1) the entity word is a noun without any quantifier.
2) the entity word is an exact subset of the caption. Do not
    modify any characters, words, and symbols.
3) the entity word should not depcit human parts, such as
    hair/leg...
4) If there are multiple same entity words, pelase also give
    the apperance of the entity word in the caption.
5) The entity word should be in singular form
6) ignore the background that appear in most of the image
    area, such as "tree", "desert", "road", "lake", "mountain
    ", "shrub", "water/waterfall.", "stream"
7) Ignore the category that belong to plants
Here are some examples, follow this format to output the
    results, note don't output the caption, give the output
    only:
### Caption: A woman in a mask and coat, with long brown hair
    , shows a small green-capped bottle to the camera in the
    street.
### {'human': ['woman'], 'subject': ['mask', 'coat', 'green-
    capped bottle']}

### Caption: A woman in a pink dress and a woman in a light
    blue shirt are seated at the bar.
### {'human': ['woman in a pink dress', 'woman in a light
    blue shirt'], 'subject': ['dress', 'shirt']}

### Caption: "runners in the court".
### {'human': [""], 'subject': []}
```

### Keyword Extraction Prompts - Second Round

**User Prompt**

```
    Given an entity word list, please reorgnize the entity
        list you output before. Do not modify any characters,
        words, and symbols of the entity word.
[Definition of humans] human that appear in the image.
[Definition of subjects] belong products(e.g. bike/cloth...)/
    animal/IP Character,
Entity words that meet the following conditions should be
    deleted.
1) human parts, such as hair/leg...
2) entity word in the plural form
3) the background spaces that appear in most of the image
    area, such as "living room/court"
4) natural landscape/plants, e.g. "tree", "desert"
Here are some examples, the output results should in json
    format:
### {'human': ['women'], 'subject': ['tops', 'pants', 'door
    ']}
{'human': [''], 'subject': []}
```

```
### {'human': ['fish'], 'subject': ['coral reef', 'coral', '
    fish']}
{'human': [''], 'subject': ['fish', 'coral', 'fish']}

### {'human': ['woman with long, flowing black hair', 'woman
    '], 'subject': ['red and silver earring', 'white garment
    ', 'traditional building', 'wooden pillars', 'tiled roof
    ']}
{'human': ['woman with long, flowing black hair', 'woman'], '
    subject': ['red and silver earring', 'white garment']}

### {'human': ['man in a black suit and white gloves', 'man
    in a black shirt and pants', 'audience'], 'subject': ['
    black suit', 'white gloves', 'green pool table', 'cue
    stick', 'shot', 'cue ball']}
{'human': ['man in a black suit and white gloves', 'man in a
    black shirt and pants'], 'subject': ['black suit', 'white
     gloves', 'green pool table', 'cue stick', 'cue ball']}
```

### 6.3.2 VISUAL GROUNDING

We adopt the vision-language model Qwen2.5-VL 7B for open-set grounding.

**Visual Grounding Prompts**

**User Prompt**

```
Outline the position of {}. If there is only one bbox
    detected, output the coordinates in JSON format. For
    example:
```json
    [
        {{"bbox_2d": [654, 103, 710, 225], "label": "man"}}
    ]
```
Otherwise, if there are multiple objects, directly return 0.
```

### 6.3.3 VISUAL SEMANTIC RECHECK

We futher use InternVL2.5 8B to filter unsuitable subjects.

**Visual Semantic Recheck Prompts**

**User Prompt**

```
Is the object in the image a valid object? The following
    conditions need to be met:
1. Specificity: The object is unique enough that it can be
    identified.
2. Completeness: a) For non-person cls, the completeness of
    the object should be above 80%. The object is mostly
    unoccluded and shown in its entirety. b) For person class
    , if the face appears, then the object can be considered
    as complete object.
3. Classes: The object should not be a) hair/leg/hand b)the
    word in the plural form. c) the background spaces that
```

```
    appear in most of the image area, such as sky. d) natural
      landscape/plants, e.g. "tree", "desert"
4. image-text consistency: the image depicts {}.
If the above conditions are met, return 1. Otherwise, return
    0. The output should only be 1/0.
```

### 6.3.4  QUERY PRODUCT

We futher use InternVL2.5 8B to find common product.

**Query Product Prompt**

**User Prompt**

```
[Task Instructions]
Strictly analyze whether the input image belongs to easily
    retrievable product photography. Output yes ONLY if both
    core criteria are satisfied; otherwise output no:
Brand Logo Visibility – The image clearly contains
    recognizable commercial brand logos (e.g., Nike/Adidas/
    Gucci/Apple) without occlusion or excessive blurring.
Complete Product Presentation – The product is fully
    displayed (no critical parts missing/folded/occluded),
    occupies the central area of the image, and maintains
    proper composition.
[Output Specifications]
Output lowercase yes/no with reason.
```

### 6.3.5  VLM VERIFICATION

We observed the limited performance of the open-sourced Qwen2.5 7B and InternVL2.5 8B models on this task, and therefore chose Doubao-1.5-Vision-Pro-32K to perform this task.

**VLM Verification Prompts**

**User Prompt**

```
person_prompt = """
Task: Verify if the main object in two images meets the
    following consistency criteria.
Output: Return "yes" if all conditions are satisfied;
    otherwise, return "no".

---
Verification Steps
1. Main Object Check
  - The main object in both images is a human.
2. Clothing Consistency
  - The humans are wearing the same clothing (color, style,
      and accessories).
3. Identity Match
  - The humans are the same person (facial features and
      physical traits match).
4. Background Difference
```

```
    - The humans are not in a similar environments or
      backgrounds.

---
Output Logic
- If all conditions are met: Return "yes".
- If any condition fails: Return "no".

The output should be "yes" or "no" only, without any other
    text or explanation.
"""

person_prompt_doubao = """
Task: Verify if the main object in two images meets the
    following consistency criteria.
Output: Return "yes" if all conditions are satisfied;
    otherwise, return "no".

---
Verification Steps
1. Main Object Check
  - The main object in both images is a human.
2. Clothing Consistency
  - The humans are wearing the same clothing (color, style,
      and accessories).
3. Identity Match
  - The humans are the same person (facial features and
      physical traits match).
4. Background Class Difference
  - The background class should be different. Please note
      this is very important. The view differece should be
      considered as the same background class type
---
Output Logic
- If all conditions are met: Return "yes".
- If any condition fails: Return "no".

The output should be "yes" or "no" only, without any other
    text or explanation.
"""

ip_animal_prompt = """
Task: Verify if the main object in two images meets the
    following consistency criteria.
Output: Return "yes" if all conditions are satisfied;
    otherwise, return "no".

---
Verification Steps
1. Main Object Check
  - The main object in both images is a ip character or
      animal. Not a food.
2. Clothing Consistency
  - The main objects are wearing the same clothing (color,
      style, and accessories).
3. Identity Match
```

```
  - The main objects are with the same identity.
4. Background Difference
  - The main objects are in different environments or
     backgrounds.
---
Output Logic
- If all conditions are met: Return "yes".
- If any condition fails: Return "no".

The output should be "yes" or "no" only, without any other
    text or explanation.
"""
```

## 6.4 BROADER IMPACT

Our work aims to enhance identity consistency and contextual diversity in subject-to-video generation, thereby improving the controllability and realism of AI-generated content. This technology holds promise for a wide range of applications, including personalized media creation, digital asset generation, and educational or entertainment content production.

Nonetheless, we recognize the potential social risks associated with realistic identity-preserving video synthesis. Such capabilities may be misused for malicious purposes, including the creation of deepfakes, impersonation, or the dissemination of misinformation. We therefore emphasize the importance of responsible research and deployment practices. In particular, we encourage the use of watermarking, provenance tracking, and informed consent mechanisms to ensure ethical and transparent use—especially in scenarios involving human likeness or identity-sensitive content.

## 6.5 THE USE OF LARGE LANGUAGE MODELS

We used a large language model solely for writing polishing: improving grammar, clarity, and stylistic consistency of sentences written originally by the authors. The model was not used to generate research ideas, experimental designs, analyses, results, or related work content.

