# OpenReview forum: "Phantom-Data:  Towards a General Subject-Consistent Video Generation Dataset"
_ICLR.cc/2026/Conference — ICLR 2026 Poster_

### Official Review · Reviewer_czCa · 2025-10-15

**Soundness:** 3
**Presentation:** 3
**Contribution:** 4
**Rating:** 10
**Confidence:** 4

**Summary:**

The paper tackles a central pain point in subject-consistent video generation—the “copy–paste” failure mode that emerges from in-pair training, where a reference frame and target video come from the same clip and the model inadvertently binds identity to background/pose. The authors introduce **Phantom-Data**, a large-scale **cross-pair** dataset (1M identity-consistent pairs across humans, animals, products, IP characters), and a three-stage pipeline to build it: (1) S2V detection via open-vocabulary grounding with quality checks (completeness/specificity/text match), (2) large-scale retrieval from **~53M videos + 3B images** with category-specialized encoders, and (3) prior-guided identity verification (e.g., logo presence for products; same long video for living beings) plus a VLM check to ensure both identity match and context diversity (Fig. 4, p.5).

Using Phantom-Data to train an open-source subject-consistent model (Phantom-wan), the paper reports clear gains in **prompt following** and **video quality** while keeping **identity consistency** comparable to in-pair baselines (Table 2, p.7). Qualitative examples (Fig. 5, p.8), ablations on subject diversity/scale (Tables 3–4, p.9), and a user study (Table 5, p.15) corroborate the improvements and the importance of cross-pair diversity.

**Strengths:**

* **Addresses a key failure mode.** Clearly identifies and targets the copy–paste issue caused by in-pair supervision, and proposes a principled data solution rather than ad-hoc augmentations (Fig. 2, p.3; Sec. 3).
* **Well-designed, scalable pipeline.** The three-stage pipeline (detection → cross-context retrieval → prior-guided verification) is systematic, with concrete checks (completeness/specificity; upper/lower similarity thresholds; VLM verification) that materially reduce false positives (Fig. 4–6, pp.5–9).
* **General beyond faces.** Moves cross-pair training from face-only domains to **general subjects** (humans, animals, products, multi-subject scenes), aligning with real user inputs (Fig. 3(d–e), p.4; Table 1, p.3).
* **Fair, controlled comparisons.** Same base model, objective, resolution, and inference settings across training regimes; metrics cover text alignment (Reward-TA), identity (DINO/GPT-4o scores), and video quality (VBench), with both quant and qual results (Sec. 4, Table 2, Fig. 5).
* **Clear, strong empirical gains.** Cross-pair training with Phantom-Data improves prompt alignment and overall video quality, with identity consistency on par with in-pair training (Table 2); ablations show benefits from subject diversity and scaling from 100k → 1M pairs (Tables 3–4).

**Weaknesses:**

* **Ambiguity around the “Face Cross-pair” baseline (Table 2).** The paper states this baseline “utilizes face-level identity matching across videos,” but the exact construction is unclear relative to **Ours**. Did “Face Cross-pair” (i) rely solely on ArcFace-based retrieval without the **prior-guided verification** step, (ii) omit **clothing/body** features for people, and/or (iii) exclude **non-face** subjects entirely? Clarifying the settings would explain the large Reward-TA gap (3.022 vs. 3.827) and the differences in DINO/GPT-4o scores (Table 2, p.7).

**Questions:**

None

---

> ### Author Response · Authors · 2025-11-20
>
> We sincerely appreciate the reviewer's thoughtful and constructive feedback. We address each concern point-by-point below.
>
> **W1: Clarification on the "Face Cross-pair" baseline construction.**
>
> A:  This “Face Cross-pair” baseline was trained  (iii) exclusively on cross-pair face data, effectively excluding all non-face subject categories (e.g., animals, products, IP characters, and full-body contexts without clear face visibility). This structural difference  explains the performance gaps observed in Table 2:
>
> - Subject Consistency (DINO/GPT-4o): Our evaluation test suite consists of  diverse cases covering humans, animals, products, and scenes. Since the "Face Cross-pair" model was never exposed to non-face categories during training, it naturally fails to maintain identity for these general subjects, resulting in lower overall consistency scores.
>
> - Prompt Following (Reward-TA): The significant lead of "Ours" (3.827 vs. 3.022) highlights that a narrow, face-centric dataset limits the model's ability to understand and execute complex, general-domain instructions. This demonstrates that to achieve both SOTA prompt following and robust subject consistency, a large-scale, diverse cross-pair dataset  is indispensable.

---

### Official Review · Reviewer_zkPr · 2025-10-30

**Soundness:** 3
**Presentation:** 3
**Contribution:** 4
**Rating:** 6
**Confidence:** 4

**Summary:**

Phantom-Data is a large-scale, general-purpose dataset designed to address the "copy-paste" problem in subject-to-video (S2V) generation. Existing models often fail to follow textual prompts because they are trained on in-pair data (reference and target frames from the same video), leading them to copy irrelevant background details from the reference image. Phantom-Data mitigates this by providing "cross-pair" data: identity-consistent reference and target frames from different videos or contexts.
Experiments using the Phantom-wan model show that training with Phantom-Data significantly improves text-prompt alignment and video quality compared to in-pair training baselines, while maintaining high identity consistency.

**Strengths:**

1. The "copy-paste" problem is a major, well-known limitation in current open-source video generation models. Tackling this via better data is a highly practical and valuable approach.

2 Unlike previous cross-pair datasets that mostly focused on faces, this dataset covers general objects (products, animals, etc.) and is very large (1 million pairs)

3. The data construction pipeline is well-designed. Specifically, combining large-scale retrieval with strict VLM-based verification  is a smart way to automate generating high-quality pairs without human annotation.

**Weaknesses:**

1. While the dataset is a great contribution, the pipeline requires an enormous pool of data (53M videos, 3B images) to work effectively. This scale is likely out of reach for most academic labs to replicate or extend on their own.

2. The experimental results primarily compare different custom data setups (in-pair, in-pair + aug, face cross-pair) against their proposed method. There is no comparison against models trained on currently existing public datasets to provide a true external baseline.

3. While quantitative metrics are provided for the different data ablation setups (Table 3), visual examples comparing the outputs of these specific ablations (e.g., "face only" vs. "+ product" vs. "+ multi-subject") would strengthen the argument for the necessity of each data component.

**Questions:**

1. Table 1 states the dataset is "Publicly Available". Does this mean the full 1 million pairs (images and corresponding video clips) will be directly downloadable, or will it be a list of URLs/IDs that users need to scrape themselves?

2. Regarding the "same long-form video" (L339) constraint for living beings: Did you experiment with trying to find the same human/animal across completely different videos?

3. What is the estimated error rate of the final VLM verification step? Are there many valid cross-pairs that get thrown out because the VLM is too conservative?

---

> ### Author Response · Authors · 2025-11-20
>
> We sincerely appreciate the reviewer's thoughtful and constructive feedback. We address each concern point-by-point below.
>
> **W1: Scale and Reproducibility for Academic Labs.**
>
> A: We thank the reviewer for acknowledging the scale of our work. We believe that bridging this gap between industrial-scale data resources and academic accessibility is precisely the core contribution of Phantom-Data.
>
> As the reviewer correctly noted, accessing and processing such a massive raw data pool (53M videos, 3B images  is indeed out of reach for most academic labs. Precisely for this reason, we undertook this heavy data engineering effort to distill these vast sources into 1 million high-quality, identity-consistent cross-pair samples.
>
> By publicly releasing this curated dataset , our goal is to provide a robust, ready-to-use resource for the community. This allows academic researchers to bypass the prohibitive data engineering bottleneck and focus their resources on algorithmic innovation to solve the "copy-paste" problem, democratizing access to high-quality S2V training infrastructure.
> ***
>
> **W2: external baseline**
>
> We would like to clarify that at the time of our submission, there were no widely adopted and publicly available large-scale datasets specifically designed for general subject-consistent video generation (covering diverse categories beyond just human faces).
>
> Regarding the recent OpenS2V dataset (noted by Reviewer#Qtiu ), we acknowledge it as a significant recent work. We will strictly include a detailed discussion and comparison with such emerging benchmarks in the final version to provide a more comprehensive external baseline.
>
> ***
> **W3: Visual examples for ablation studies.**
>
> A:  We will include a dedicated figure in the Appendix of the final version, explicitly showcasing the visual differences between models trained on "face only" data versus those enriched with "product" and "multi-subject" data.
>
> ***
>
> **Q1: Data Availability (URLs vs. Download).**
>
> A: To strictly adhere to copyright regulations and data compliance standards, we will release the dataset in the form of URLs and metadata IDs, rather than hosting the raw video files directly. This practice aligns with other large-scale open-source datasets (e.g., LAION, Koala-36M) to ensure the dataset's long-term legality and availability.
>
> ***
>
> **Q2: "same long-form video" constraint for living beings**
>
> A:  We did explore global retrieval for living beings during our preliminary experiments but decided to enforce the "same long-form video" constraint based on two key findings:
>
> 1. Sufficient Diversity: We found that sampling from different segments of a long-form video already provides sufficient contextual diversity to break the "copy-paste" pattern. As illustrated in Figure 6(a)()()()(), the visual context (lighting, background, pose) varies significantly across minute-level intervals within the same video, effectively serving our training needs without the risks of global retrieval.
>
> 2. Controlling False Positives: Global retrieval for specific living beings (especially animals or non-celebrity humans) introduces an extremely high false positive rate. Without the strong prior of "co-occurrence in the same footage," distinguishing between the exact same individual and a look-alike (e.g., two similar Golden Retrievers) is incredibly challenging, even for advanced VLMs. The "same long-form video" constraint acts as a necessary anchor to ensure identity precision.
>
> ***
> **Q3: Estimated error rate and VLM conservativeness.**
>
> A: To quantitatively estimate the error rate, we conducted a manual inspection on 200 randomly sampled pairs from the final dataset. Our evaluation indicates a false positive rate of approximately 10.5%, suggesting that the majority of the retained pairs are identity-consistent.

---

### Official Review · Reviewer_EDtH · 2025-10-31

**Soundness:** 3
**Presentation:** 3
**Contribution:** 3
**Rating:** 4
**Confidence:** 3

**Summary:**

This paper introduces Phantom-Data, a cross-pair subject-to-video consistency dataset for video personalization, comprising one billion identity-consistent pairs across diverse categories. Experiments indicate the high quality and potential impact of the dataset.

**Strengths:**

- The paper presents a well-designed pipeline for curating subject-video paired datasets, leveraging Visual Language Models (VLMs) to achieve robust subject-attribute pairing.
- The dataset is large-scale, providing valuable resources for advancing research in video personalization.
- The curated data effectively addresses the prevalent copy-paste issue encountered in video personalization tasks.

**Weaknesses:**

- It remains unclear how much the VLM-based pipeline improves over simpler approaches, such as using GPT-4o for generating image variations. As shown in Figure 7, though having drawbacks, generative models might sometimes even offer advantages in achieving controllable variations. Additionally, potential artifacts from generative approaches could be mitigated through VLM-driven verification and filtering, which the paper does not explore.
- The evaluation is somewhat limited, as the dataset is tested exclusively with Wan2.1. Given that many established methods in video personalization predate the Wan series, broader evaluation across multiple models would strengthen the paper’s claims and demonstrate the wider utility of the dataset.

**Questions:**

n/a

---

> ### Author Response · Authors · 2025-11-20
>
> We sincerely appreciate the reviewer's thoughtful and constructive feedback. We address each concern point-by-point below
>
> **W1: VLM-based Real Data Pipeline vs. Synthetic Data Generation.**
>
> A :  We agree that synthetic data offers unique advantages in controllability and editability. However, we chose the real-world retrieval pipeline for three critical reasons:
>
> 1. Scalability and Cost Constraints: Generating high-quality, consistent pairs at the scale of Phantom-Data (1 million pairs) is prohibitively expensive. For instance, recent work like OpenS2V notes that due to resource constraints, they could only select the top 10k samples to construct GPT-frame pairs out of a larger candidate pool. In contrast, our retrieval approach leverages existing large-scale video corpora (53M+ videos), allowing us to scale up to 1 million high-quality pairs efficiently without the immense computational and API costs associated with generating millions of image variations.
>
> 2. Identity Consistency Upper Bound: While VLM-driven verification can filter out bad samples, the quality of the dataset is ultimately bounded by the generator's capability. Even state-of-the-art subject-consistent models (e.g., OmniGen, Nano-Banana) struggle to preserve intricate details (e.g., complex clothing textures) at a "real-image level." As shown in Figure 7 of our paper， generative models often introduce subtle identity shifts or artifacts. Relying on synthetic data would inherently bake these imperfections into the training set, whereas real-world retrieval guarantees that the subject is physically the same, shifting the challenge solely to finding the right frames.
>
> 3. Real-World Diversity vs. "Imagined" Prompts: Synthetic data is limited by the diversity of prompts and the generative model's training distribution. It is difficult to manually design prompts that cover the vast complexity of the physical world—such as unpredictable lighting conditions, motion blur, complex occlusions, and chaotic backgrounds. Our retrieval pipeline naturally mines these real-world variations, which are crucial for training a robust model that generalizes well to user-uploaded photos in diverse, unconstrained environments.
>
> While we prioritized real-world retrieval in this work to establish a robust baseline for realism and consistency, we fully recognize the complementary value of synthetic data, particularly for tasks requiring high editability (where real-world data may lack granular control). We appreciate the reviewer's insight and plan to investigate hybrid approaches in future work—potentially using synthetic data in post-training stages to enhance the model's editability capabilities.
>
> ***
>
> **W2:  Effectiveness on other open-source frameworks**
>
> A: We agree that evaluating on diverse architectures is crucial to verify the generalization of our dataset. we conducted additional experiments on CogVideoX-5B and HunyuanVideo-14B.  The results, presented in the tables below, demonstrate that Phantom-Data consistently generalizes across different models:
>
> 1. Significant Gains in Text Alignment & Video Quality: Training with Phantom-Data ("Ours") yields substantial improvements in Prompt Following (Reward-TA) and Video Quality metrics (e.g., Motion, IQ, Background consistency) compared to the In-pair baseline on both frameworks.
>
> 2. Competitive Identity Preservation: While pixel-level metrics like DINO show a slight trade-off (similar to our observation with Wan2.1, as cross-pair training reduces overfitting to the exact pixel layout of the reference), the GPT-4o scores—which align better with human perception of identity—are significantly higher. This confirms our dataset maintains robust semantic identity consistency.
>
> For CogVideoX-5B:
> | Methods | DINO ↑ | GPT-4o ↑ | Reward-TA ↑ | Temporal ↑ | Motion ↑ | IQ ↑ | BG ↑ | Subj ↑ |
> | :--- | :---: | :---: | :---: | :---: | :---: | :---: | :---: | :---: |
> | In-pair | **0.453** | 2.327 | 1.932 | 0.950 | 0.977 | 0.713 | 0.905 | 0.942 |
> | **Ours** | 0.421 | **3.122** | **3.653** | **0.955** | **0.980** | **0.719** | **0.953** | **0.971** |
>
> For HunyuanVideo-14B
> | Methods | DINO ↑ | GPT-4o ↑ | Reward-TA ↑ | Temporal ↑ | Motion ↑ | IQ ↑ | BG ↑ | Subj ↑ |
> | :--- | :---: | :---: | :---: | :---: | :---: | :---: | :---: | :---: |
> | In-pair | **0.393** | 2.103 | 2.132 | 0.991 | 0.987 | 0.721 | 0.911 | 0.932 |
> | **Ours** | 0.355 | **2.923** | **3.953** | **0.993** | **0.994** | **0.753** | **0.962** | **0.951** |
> ***

---

### Official Review · Reviewer_Qtiu · 2025-11-01

**Soundness:** 3
**Presentation:** 3
**Contribution:** 2
**Rating:** 4
**Confidence:** 4

**Summary:**

This paper introduces Phantom-Data, a large-scale cross-pair dataset for subject-consistent video generation, aiming to mitigate the copy-paste problem caused by in-pair training. The dataset is built via a three-stage pipeline combining open-vocabulary detection, cross-context retrieval, and prior-based identity verification. Experiments show improved prompt alignment and video quality while maintaining subject consistency.

**Strengths:**

1. The motivation is clear and addresses a real issue (“copy-paste”) in subject-consistent video generation.
2. The dataset construction pipeline is well-designed, integrating VLM-based detection and verification modules.
3. The ablation and user study provide some evidence of effectiveness.

**Weaknesses:**

1. The prompts used in the dataset are overly simple and sparse, failing to accurately describe video semantics or capture complex spatiotemporal relations. This limits the dataset’s potential to improve text-video alignment in realistic scenarios.
2. The paper evaluates only on the Phantom-Wan model. Without experiments on other open-source frameworks (e.g., CogVideoX, VACE, HunyuanVideo), it’s hard to judge whether the dataset quality generalizes.
3. The paper compares mainly with older datasets, while recent large-scale subject-consistent datasets are not included. It is unclear how Phantom-Data performs relative to the latest baselines, such as OpenS2V.

**Questions:**

As seen in weakness

---

> ### Author Response · Authors · 2025-11-20
>
> We sincerely appreciate the reviewer's thoughtful and constructive feedback. We address each concern point-by-point below
>
> **W1:  Simple text prompt**
>
> A: We would like to clarify that the video captions in our dataset are  dense and semantically rich. The "simple and sparse" prompts observed in the qualitative results (e.g., Fig. 1 and Fig. 5) were manually simplified solely for visualization purposes. Due to space constraints in the main paper, we condensed the prompts to ensure the figures were legible and to direct the readers' focus to the core visual difference.
>
> ***
> **W2:  Effectiveness on other open-source frameworks**
>
> A: We agree that evaluating on diverse architectures is crucial to verify the generalization of our dataset. Regarding the specific frameworks mentioned, we note that VACE shares the same underlying architecture as our current baseline model (Wan2.1). Therefore, to strictly test the dataset's robustness across distinct architectural designs, we prioritized conducting additional experiments on CogVideoX-5B and HunyuanVideo-14B.  The results, presented in the tables below, demonstrate that Phantom-Data consistently generalizes across different models:
> 1. Significant Gains in Text Alignment & Video Quality: Training with Phantom-Data ("Ours") yields substantial improvements in Prompt Following (Reward-TA) and Video Quality metrics (e.g., Motion, IQ, Background consistency) compared to the In-pair baseline on both frameworks.
> 2. Competitive Identity Preservation: While pixel-level metrics like DINO show a slight trade-off (similar to our observation with Wan2.1, as cross-pair training reduces overfitting to the exact pixel layout of the reference), the GPT-4o scores—which align better with human perception of identity—are significantly higher. This confirms our dataset maintains robust semantic identity consistency.
>
> For CogVideoX-5B:
> | Methods | DINO ↑ | GPT-4o ↑ | Reward-TA ↑ | Temporal ↑ | Motion ↑ | IQ ↑ | BG ↑ | Subj ↑ |
> | :--- | :---: | :---: | :---: | :---: | :---: | :---: | :---: | :---: |
> | In-pair | **0.453** | 2.327 | 1.932 | 0.950 | 0.977 | 0.713 | 0.905 | 0.942 |
> | **Ours** | 0.421 | **3.122** | **3.653** | **0.955** | **0.980** | **0.719** | **0.953** | **0.971** |
>
> For HunyuanVideo-14B
> | Methods | DINO ↑ | GPT-4o ↑ | Reward-TA ↑ | Temporal ↑ | Motion ↑ | IQ ↑ | BG ↑ | Subj ↑ |
> | :--- | :---: | :---: | :---: | :---: | :---: | :---: | :---: | :---: |
> | In-pair | **0.393** | 2.103 | 2.132 | 0.991 | 0.987 | 0.721 | 0.911 | 0.932 |
> | **Ours** | 0.355 | **2.923** | **3.953** | **0.993** | **0.994** | **0.753** | **0.962** | **0.951** |
> ***
> **W3: Comparison with recent baselines (e.g., OpenS2V).**
>
> A: We thank the reviewer for highlighting OpenS2V. Due to the significant time required to download and process the OpenS2V dataset (estimated at over three weeks), it was not feasible to conduct a full retraining and evaluation cycle within the rebuttal period.  Therefore, we wish to highlight the core methodological distinction regarding Data Construction that differentiates Phantom-Data from approaches like OpenS2V:
>
> *Synthetic Generation (OpenS2V) vs. Real-World Retrieval (Ours).* While both methods aim to solve the copy-paste problem by providing diverse context, their approaches are fundamentally different. OpenS2V relies heavily on "Nexus Data" (e.g., GPT-Frame pairs), where reference images are synthesized by generative models to create cross-view consistency. In contrast, Phantom-Data is constructed through large-scale retrieval from real-world videos and images. This real-world approach offers three key advantages:
>   1. Real-World Diversity: Our data captures the complex, natural variations in lighting, texture, motion blur, and background clutter found in reality  Synthetic data, while consistent, often lacks these subtle imperfections and environmental complexities.
>   2. Training-Inference Consistency: In real-world applications, users typically provide actual photos (e.g., a selfie or a product shot) rather than synthetic images. Training on real-world retrieval data ensures the model is optimized for this realistic input distribution, minimizing the domain gap.
>   3. Scalability: Generating million-scale synthetic datasets incurs prohibitive API costs. In contrast, our retrieval pipeline leverages existing large-scale video corpora, making it a highly scalable and cost-effective solution for constructing general-purpose datasets.

---

### Meta-Review · Area_Chair_5m7f · 2026-01-03

**Summary:**

There are mixed reviews for this paper with two accept (10 and 6) and two rejects (4 and 4)
Since 10 is very out of range compared with the other three, I won't take reviewer with 10 into my final consideration. My recommendation is mainly based on the other three reviewers:


Main motivation for this paper is a proposed Subject-Consistent Video Generation Dataset
Here are the main concerns:
1. The dataset's prompt are not dense enough which may/will hurt the  final model performance.
2. The proposed dataset  is only eval on a certain model but not many video-generation models.


The rebuttal in general are not strong compared with other papers but authors do address reviewer's concerns.

**Reviewer Concerns:**

Mainly addressed by authors and I don't see any major outstanding one

**Reviewer Scores:**

There are mixed reviews for this paper with two accept (10 and 6) and two rejects (4 and 4)

---

### Decision · Program_Chairs · 2026-01-26

Accept (Poster)